# Transcriptome Analysis Unravels Metabolic and Molecular Pathways Related to Fruit Sac Granulation in a Late-Ripening Navel Orange (*Citrus sinensis* Osbeck)

**DOI:** 10.3390/plants9010095

**Published:** 2020-01-11

**Authors:** Li-Ming Wu, Ce Wang, Li-Gang He, Zhi-Jing Wang, Zhu Tong, Fang Song, Jun-Fan Tu, Wen-Ming Qiu, Ji-Hong Liu, Ying-Chun Jiang, Shu-Ang Peng

**Affiliations:** 1Key Laboratory of Horticultural Plant Biology, Ministry of Education, College of Horticulture and Forestry Sciences, Huazhong Agricultural University, Wuhan 430070, China; wuliming2005@126.com; 2Research Institute of Fruit and Tea, Hubei Academy of Agricultural Science, Wuhan 430064, China; hysc1230@sohu.com (C.W.); lghe44@aliyun.com (L.-G.H.); wzjjsz@sohu.com (Z.-J.W.); tsady@163.com (Z.T.); fsong_ray@163.com (F.S.); tujunfan2005@126.com (J.-F.T.); qiuwm1984@sina.com (W.-M.Q.)

**Keywords:** late-ripening navel orange, low temperature, juice sac granulation, physiological biochemistry, transcript

## Abstract

Lanelate navel orange (*Citrus sinensis* Osbeck) is a late-ripening citrus cultivar increasingly planted in China. The physiological disorder juice sac granulation often occurs in the fruit before harvest, but the physiological and molecular mechanisms underlying this disorder remain elusive. In this study, we found that fruit granulation of the late-ripening navel orange in the Three Gorges area is mainly caused by the low winter temperature in high altitude areas. Besides, dynamic changes of water content in the fruit after freezing were clarified. The granulation of fruit juice sacs resulted in increases in cell wall cellulose and decreases in soluble solid content, and the cells gradually became shrivelled and hollow. Meanwhile, the contents of pectin, cellulose, and lignin in juice sac increased with increasing degrees of fruit granulation. The activities of pectin methylesterase (PME) and the antioxidant enzymes peroxidase (POD), superoxide dismutase, and catalase increased, while those of polygalacturonase (PG) and cellulose (CL) decreased. Furthermore, a total of 903 differentially expressed genes were identified in the granulated fruit as compared with non-disordered fruit using RNA-sequencing, most of which were enriched in nine metabolic pathways, and qRT-PCR results suggested that the juice sac granulation is closely related to cell wall metabolism. In addition, the expression of *PME* involved in pectin decomposition was up-regulated, while that of *PG* was down-regulated. Phenylalanine ammonia lyase (*PAL*), cinnamol dehydrogenase (*CAD*), and *POD* related to lignin synthesis were up-regulated, while *CL* involved in cellulose decomposition was down-regulated. The expression patterns of these genes were in line with those observed in low-temperature treatment as revealed by qRT-PCR, further confirming that low winter temperature is associated with the fruit granulation of late-ripening citrus. Accordingly, low temperature would aggravate the granulation by affecting cell wall metabolism of late-ripening citrus fruit.

## 1. Introduction

Late-ripening citrus (*Citrus sinensis* Osbeck) needs overwintering cultivation due to its late natural maturation. Its fruit growth period is more than 11 months, which is an important trait to optimize the structure of citrus varieties, as well as to prolong the maturation and achieve a year-round fresh fruit supply. The Three Gorges Reservoir is the most suitable area for the planting of late-ripening citrus in China. For example, Lanelate navel orange planted in this area is a superior cultivar with excellent fruit quality [1]. Freezing injury should be avoided for late-ripening varieties so that the fruit can survive the winter safely. However, in commercial production, the late-ripening variety is vulnerable to the influence of low temperature in winter and the fruits are prone to granulation. Overwintering fruits in nature are easily affected by low temperature in winter [2], and may show severe granulation symptoms at the flowering and shoot growth stages of the following year, but the role of temperature in fruit granulation mechanism is still unknown.

Fruit granulation is a physiological disorder of citrus, especially sweet orange. It was first reported in California by Bartholomew [3] and then studied in Australia, Brazil, China, India, Israel, and Japan [4,5,6,7,8,9,10]. Granulation of citrus fruit is associated with the loss of water in the juice sacs, leaving behind hard, dry, shrunken, and grey-white juice sacs without extractable juice [9,11,12], which severely affects the internal structure, flavor, sugar, and acid quality of the fruit [11]. Previous studies have demonstrated that citrus juice sac granulation is often correlated with cell wall thickening and lignin deposition [6,13], and changes in cell wall structure of juice vesicles were suggested to be associated with the increasing contents of cell wall components including pectin, lignin, hemicellulose, and cellulose, and the increase in lignin content of juice sacs plays a significant role in citrus fruit granulation process [14,15,16,17]. Some studies have also reported an association of particular enzymes with citrus granulation. These enzymes include peroxidise (POD), superoxide dismutase (SOD), polyphenol oxidase (PPO), polygalacturonase (PG), and pectin methylesterase (PME) [8,11,18].

Fruit granulation causes decreases in juice percentage, as well as in water, soluble solid and titratable acid contents, resulting in a lighter color of the juice sac and a severe decline of the intrinsic flavor quality of the fruit [9,12,13]. In the Three Gorges Reservoir area of Hubei Province, the late-ripening overwintering navel orange is frequently faced with the problem of fruit granulation. However, there have been no reports on juice sac granulation caused by freezing in late-ripening citrus fruit during on-the-plant development. The causes of fruit granulation may be quite different among varieties with different maturation periods, and the changes in physiology and expression of genes related to fruit granulation remain elusive.

In this study, late-ripening navel oranges with granulated and non-disordered juice sacs were selected as the experimental materials. First, the morphological structures of fruit and juice sacs with different levels of low water content were analyzed. Physiological indicators related to granulation were determined, including the contents of water, sugars, citric acids, cell wall components such as pectin, cellulose, and lignin, and the activities of related enzymes, such as PME, PG, cellulase (CL), POD, SOD, and catalase (CAT). Then, the orange genome [19] was used to annotate the transcriptome data, and RNA-sequencing [20,21] was performed to screen the differentially expressed genes (DEGs) of the granulated versus non-disordered fruit. Besides, the key genes related to cell wall metabolism were screened. The changes in gene expression of enzymes related to juice sac granulation were further analyzed by RT-PCR to elucidate the mechanism of granulation in fruit.

## 2. Results

### 2.1. Occurrence of Granulation in Late-Ripening Navel Orange Fruit

The relationship of fruit granulation with low winter temperature of different years (2011–2018) in Zigui County of Three Gorges Reservoir Area was analyzed (Appendix A). Considering that the altitude is related to the temperature, the average monthly temperature in winter and the extremely low temperature in January and February (Appendix A) were measured by placing an automatic recorder of temperature and humidity in orchards at different altitudes. Statistics showed that with increasing altitude, the average monthly temperature decreased. At the altitude of 520 m, the average monthly temperature in the orchards dropped to 3.8 °C in February, and the extremely low temperature was −4.1 °C. Over the 11 years, fruit granulation in late-ripening navel orange was negatively correlated with freezing temperatures in January, although it was positively correlated with the number of low temperature days in January.

We then investigated the fruit granulation rate (Appendix A) and changes in fruit juice sac’s water content (Figure 1A) during the fruit ripening period (235 days after flowering (DAF) to 400 DAF) at different altitudes (520 m, 380 m, 220 m) in Zigui County of the Three Gorges Reservoir. Low winter temperature at high altitude areas (520 m) was associated with fruit granulation, and non-disordered fruit was observed in lower altitude areas (380 m and 220 m) (Appendix A). We further clarified the changes in water content of fruit remained on the trees after freezing, and found that the water content of fruit at 520 m altitude was lower at the later stage than at the earlier stage (Figure 1A). Fruit freezing damage was observed in the orchards at 520 m altitude, with a small part of fruit peel showing freezing damage spots and oil gland collapse, and fruit falling was also observed in late February (300 DAF) (data not shown).

No fruit granulation was detected at 235 DAF in late-ripening citrus orchards at 520 m altitude in the Three Gorges area (Figure 1B). In contrast, a granulation incidence of 5% was found at 335 DAF, which was increased by 3- and 10-folds at 350 and 365 DAF, respectively. We further found that the occurrence of granulation in late-ripening navel orange fruit was closely related to the phenological phases of fruit trees. At the same time, the degree of fruit granulation was normal (level 0), slight (level 1), medium (level 2), and serious (level 3) in accordance with the sequential phenological phases of the fruit trees (Figure 2).

It was observed that the fruit juice sacs were healthy at 235 DAF (Figure 2A). At 335 DAF (Figure 2B), the fruit showed slight granulation, mainly in a small part of the juice sac near the central column of the fruit. At 350 DAF (Figure 2C), a moderate level of granulation was observed, which then spread from the central column to the periphery, resulting in shrinking of the juice sacs and gaps between them. At 365 DAF (Figure 2D), the fruit exhibited serious granulation with large spaces, drying out, and cavitation between sacs, resulting in shrivelled and empty sacs with a lighter color.

The histological observation results showed that PAS staining [22] was deepened with the aggravation of juice sac granulation, which resulted in increased cell wall fibres, lignin, and wall thickening (Figure 2B); decreased intracellular content; and gradually shrunken and hollow cells (Figure 2C,D).

### 2.2. Juice Sac Quality and Color at Different Granulation Levels

A comparison of the fruit quality at different granulation levels showed that there were significant decreases in fruit density (Figure 3A), water content and juice percentage (Figure 3B), the contents of soluble solids and titratable acids (Figure 3C) along with increasing degree of granulation, indicating a serious overall decline of fruit quality. In addition, fruit juice sac color became lighter with increasing granulation as denoted by decreases in red (a value) and yellow (b value) coloration (Figure 3D).

### 2.3. Pectin, Cellulose, and Lignin Content and Metabolic Enzyme Activity

The contents of pectin, cellulose, and lignin in the juice sacs of late-ripening navel oranges increased with increasing granulation, which became significantly higher than those in normal juice sacs (*p* ≤ 0.05; Figure 4A,B). The contents of pectin and lignin in severely granulated fruit juice sacs were increased by 56.6% and 37.0% respectively, and cellulose was increased by 1.6-folds. We then compared the activities of cellulase (Figure 4C), polygalacturonase (Figure 4D), and pectin methylesterase (Figure 4E) in juice sac cell walls at different granulation levels. The results showed that the activity of cellulase (Figure 4C) and polygalacturonase (Figure 4D) decreased by 50.7% and 31.0% in level 3 granulated fruit juice sacs, respectively, while that of pectin methylesterase increased significantly (Figure 4E) with increasing levels of juice sac granulation by 1.14-folds in level 3 granulation. The activities of antioxidant enzymes showed significant changes (Figure 4F): the activity of SOD and POD increased significantly, by 1.24 folds and 3.32 folds in level 3 granulation, respectively, while that of CAT first increased and then decreased.

### 2.4. Sequencing and Analysis of Digital Gene Expression Profiles

#### 2.4.1. Analysis of Late-Ripening Navel Orange Fruit Transcriptome

Transcriptomic sequence libraries of granulated and normal juice sacs were constructed (Appendix A), resulting in 11.77–12.99 million raw reads, and the high-quality reads exceeded 98.8%, with an average of 12.19 million. Each read had a length of 49 bp; more than 76% of the high-quality reads could be uniquely matched with the genome; and more than 60% of them were perfectly matched.

The total clean reads matched with 21,903 genes of the sweet orange genome (http://citrus.hzau.edu.cn/orange/). According to RPKM values, a total of 903 differentially expressed genes (DEGs) were identified (|log_2_ (RPKM ratio GR/CK)| ≥ 1, probability ≥ 0.8) from the granulated fruit samples when compared with the non-disordered fruit samples, among which 476 were up-regulated and 427 were down-regulated, and the majority log_2_ values of DEGs showed 2- to 4-fold changes (Figure 5A, Appendix A).

A total of 20 genes, which are involved in protein metabolism, sugar metabolism, hormone signal response, cell component, and other pathways, were randomly selected to verify the accuracy of the digital expression profile test. Real-time PCR showed that the expression patterns of 19 genes in granulated and normal fruit were consistent with the sequencing results of expression profiles, but with some differences in fold changes (Figure 5B). In addition, a high correlation coefficient (R^2^ = 0.7012) was obtained (Figure 5C), indicating good correlations of the data and high reliability of the sequencing results.

#### 2.4.2. Functional Annotation of Differentially Expressed Genes

In order to obtain the functional annotations of DEGs and extract their corresponding sequences, BLASTx (E-value ≤ 10^−5^) analysis based on nonredundant protein sequence (NR) in NCBI was carried out to obtain their basic annotation information with the WEGO analysis tool (http://wego.genomics.org.cn/cgi-bin/wego/index.pl). GO annotation analysis was performed on DEGs (Figure 6). The results showed that 316 up-regulated DEGs were annotated to 38 GO classification terms (second-level classification) and 279 down-regulated DEGs were annotated to 38 other GO classification items. In cell component classification, most of the gene products were located in cells and organelles. The molecular functions mainly included binding, catalysis, and transport activity. Besides, these DEGs mainly participated in the cells’ biological processes, metabolic processes, and response stimuli.

#### 2.4.3. Enrichment Analysis of Differential Gene Pathways

To reveal the differences in the metabolic pathways of DEGs, KEGG pathway enrichment analysis was performed. The results showed that for 903 DEGs, 486 were annotated in 105 pathway terms. Nine pathways were identified as significantly enriched with corrected p-values (q-value) at the ≤ 0.05 level (Figure 7). Among them, the secondary metabolite biosynthesis, tryptophan metabolism, cutin, suberine and wax biosynthesis, and plant hormone signal transduction were the most significantly enriched pathways (q-value ≥ 0.1).

### 2.5. Analysis of Granulation-Related Genes and Their Expression Changes

The RNA-sequencing data of granulated and non-disordered fruit were analyzed (Table 1). Significantly different expression was found for some genes, including the pectin methylesterase gene (Orange 1.1t00214, *Cs*2g07660), alcohol acyltransferase gene (*Cs*2g31410), auxin binding protein gene (*Cs*1g15830), polygalacturonase inhibitor protein gene (*Cs*7g18970, Cs7g01980) and calcium binding protein (*Cs*6g21420) gene, which are closely related to cell wall metabolism. The pectin methylesterase gene (*PME*) was significantly enriched in the cell wall metabolic pathway (Figure 7). The alcohol acyltransferase (*AAT*) gene was enriched in the cutin, suberine, and wax biosynthesis pathway (Figure 7). The auxin-binding protein gene (*Cs*1g15830) was enriched in the plant hormone signal transduction pathway of cell walls (Figure 7). We then carried out a quantitative expression analysis (Figure 8) of selected DEGs related to granulation in fruit with different levels of granulation. The results showed that the genes related to the cell wall metabolism in fruit juice sac were involved in fruit granulation, such as the *PME1*, *PME2*, and *AAT*. The genes showed an overall trend of increases in expression, with slight changes in the first three stages of fruit granulation, and significant increases in the later stages. The auxin binding protein gene (IAA26) and calcium binding protein gene (CABP) showed down-regulated expression in the early stage while up-regulated expression in the later stage. The expression of polygalacturonase inhibitor protein gene (PGIP1, PGIP2) was down-regulated.

### 2.6. Gene Expression Related to Cell Wall Metabolism in Granulated Fruit

Overwintering fruit suffered from different levels of granulation at different phenological stages in the following spring. The DEGs related to fruit juice sac granulation were screened by RNA-sequencing (Table 1), and the expression of genes related to cell wall metabolism in fruit juice sacs also showed significant changes accordingly (Figure 9). The expression of the pectin-related *PME* gene was significantly up-regulated with increasing fruit granulation, while that of *PG* gene and *CL* gene related to cellulose degradation was down-regulated, particularly the *CL* gene. The *PAL*, *CAD*, and *POD* genes, which are related to lignin synthesis, were significantly up-regulated.

### 2.7. Gene Expression of Metabolic Enzymes in Cell Wall of Normal Juice Sacs Treated at 4 °C

The gene expression of metabolic enzymes in cell walls of non-disordered juice sacs of late-ripening navel orange showed significant changes under low-temperature stress (Figure 10). With the extension of low-temperature treatment, the pectin-related *PME* gene and lignin synthesis-related *PAL* gene were significantly up-regulated; *PG* gene was first significantly up-regulated and then down-regulated; the *CL* gene related to cellulose degradation was significantly down-regulated; *CAD* and *POD* genes, which are associated lignin synthesis, were significantly up-regulated from 3 h, and then decreased to the initial levels in the end. The variations of the expression of these genes were basically consistent with the changes in granulation development of late-ripening navel orange.

## 3. Discussion

### 3.1. Effects of Winter Low Temperature on Fruit Granulation

Citrus fruit juice sac granulation is a serious physiological disorder occurring in pre-harvest season or post-harvest storage in different citrus species such as orange, pummelo, and grapefruit [6,14,23]. So far, the cause of the occurrence of this disorder has not been fully understood. A number of factors such as climatic factors, rootstock, fruit size, tree age, irrigation, fertilization, pruning, and delayed harvesting have been reported to be possibly associated with the disorder [8,10,11]. In our investigation, late-ripening navel orange at high altitudes was affected by winter low temperature (Appendix A). The granulation degree of the fruit remained on the tree increased gradually from slight to heavy in several stages of the following year: germination, budding, flowering, and shoot growth stages (Figure 2). The rate of fruit granulation also increased significantly. The most serious fruit granulation was observed during the flowering period (Figure 1B). We considered that juice sac granulation during fruit retention under persistent low temperatures results in injury or necrosis of some juice sac tissues, destroys the semi-permeability of the cell protoplasmic membrane, and reduces the ability of the juice sacs to hold juice and water, resulting in collapses of the sacs and gaps between them (Figure 2). Besides, the pre-harvest granulation of overwintering fruit is related to the flower and fruit development and shoot growth in the following spring. The occurrence rate and degree of fruit granulation reached the peak in the flowering and shoot growth periods. Therefore, the overwintering fruit remained on the tree may compete for water and nutrition with current-season flowers, fruits, and shoots in this phenological period.

During the citrus granulation period, the respiration intensity of the fruit and the internal physiological consumption increase, accompanied by decreases in sugars, organic acids, vitamins, and dry matter in juice cells, especially soluble sugars and organic acids [9,11,12,24,25]. In our experiment, the granulation rate and degree in the fruit remained on the tree showed obvious increases in florescence, and correspondingly the juice sac water content and juice rate decreased, leading to decreases in fruit weight and density. Meanwhile, the contents of soluble solids and titratable acids in fruit decreased significantly, and the color of the juice sac became lighter, leading to a decline of the internal quality and flavor of the fruit (Figure 3). Consequently, the quality of late-ripening citrus fruit decreased due to granulation during tree retention, which is basically consistent with the results of other citrus varieties during storage [12,18,25].

With increasing levels of fruit granulation, the fruit juice vesicle will become tough, dry, colorless, and lignified, which diminishes the nutritional and commercial values of the fruit [9,13]. The levels of lignin, pectin, hemicellulose, and cellulose all increase in granulating fruit juice sacs [14,26]. Especially, the juice sac lignin content increases during granulation and plays a vital role in this process [16,26]. Similar results were obtained in our experiment. The granulated juice sac of late-ripening navel orange contained coarse edible residues due to the higher contents of cellulose, lignin, and pectin in the juice sac (Figure 4A,B). Cellulose and lignin, the most abundant structural polymers in plant cell walls, act to protect plants from biotic and abiotic stresses [27], and mechanically support and assist the microtubule transport of water and solutes [28].

In addition, some special enzymes, such as PME, PPO, POD, and SOD, have close relationship with granulation, and some changes in the enzyme activities occur in the granulated fruit [8,10,11,18]. The researchers believe that the antioxidant enzyme activities are higher in granulated fruit than in healthy fruit. In this study, the enzymes related to cell wall metabolism were also found to be associated with the granulation of late-ripening citrus fruit (Figure 4C–F). With increasing fruit granulation, PME’s activity increased while PG’s activity decreased, which would result in the formation of more demethylate-soluble peptic acid. Thus, the increased amount of pectic acid in granulated tissues combines with calcium to form insoluble calcium pectate. CL’s activity showed a significant downward trend, resulting in a decrease in the rate of cellulose decomposition with the accumulation of more cellulose in the juice sacs and aggravation of fruit granulation, while the enzyme activities of SOD, POD, and CAT related to antioxidation increased significantly, among which POD is related to lignin formation [29]. Some carbohydrates could be converted into lignin to improve the degree of lignification.

### 3.2. Gene Expression in Late-Ripening Navel Orange Fruit under Low Winter Temperature

Transcriptome changes of sucrose and starch metabolism are related to puffing or section-drying disorder, and the gene expression related to starch output and sugar transformation is enhanced in puffing fruit as indicated by gene chip analysis [30]. Puffing mostly alters primary metabolism, such as fatty acid, pentose phosphate, and glyceride [31]. Besides, some other studies have shown that lignin metabolism is closely associated with the process of juice sac granulation [13,14,16,26]. In this study, RNA-sequencing technology was used to further confirm that the fruit granulation of late-ripening navel orange is closely related to cell wall metabolism and the related genes were identified. The expression of the DEGs greatly affected the occurrence of fruit granulation. It has been found that the development of fruit granulation during postharvest storage is closely related to cell wall metabolism, especially the metabolism of cellulose and lignin [13,24,26]. Similar results were obtained in this study of late-ripening navel orange before harvest. Changes were observed in the expression of some genes, including *PME* and *PG* genes related to pectin decomposition, *CL* gene related to cellulose decomposition, and *PAL*, *CAD* and *POD* genes related to lignin synthesis (Figure 9). Besides, the expression patterns of these genes were basically consistent with those observed under low-temperature treatment, suggesting that winter low temperature contributes to the granulation in overwintering fruit of late-ripening citrus (Figure 10).

Therefore, the mechanism of granulation development in late-ripening navel orange fruit can be summarized as below. Late-ripening citrus fruit are damaged by low temperature in winter season, with down-regulation of the polygalacturonase (*PG*) gene but up-regulation of the pectin methylesterase (*PME*) gene. Some pectin is demethylated and esterified by PME enzyme with increased solubility, but the low activity of PG enzyme may be insufficient to completely dissociate the side chains and decompose the pectin, resulting in the accumulation of higher-molecular-weight pectin and increasing the juice viscosity. The pectic substances binding with the increased levels of Ca^2+^ can cause the hardening of juice sacs [15]. When the pectin accumulation reaches a certain level, it can form gel under the action of Ca^2+^ by forming calcium bridge or being immobilized by ester bonds to trap water in the gel. At the same time, low-temperature stress causes irreversible damage to cellulase (CL), and the expression of *CL* gene is significantly down-regulated to reduce the activity of cellulose and hinder its degradation process, leading to increases in cellulose accumulation. In addition, under low-temperature stress, the *PAL, CAD*, and *POD* genes, which are related to lignin synthesis, are significantly up-regulated. With increases in lignin synthesis, the decomposition rate is lower than that of synthesis, which will lead to secondary wall growth, thickening, and granulation.

### 3.3. Differences in Fruit Granulation between Overwintering Citrus Remained on the Tree and Post-Harvest Storage Citrus

Due to the long growth period of late-ripening Navel orange, the tree hanging time of the fruit is more than 11 months. At the harvest in the late spring and early summer of the second year, new flowers and old fruit remained on the same tree, and the causes of pre-harvest fruit granulation should be quite different from those of early and middle-ripening varieties. We considered that the fruit granulation before the harvest of late-ripening citrus in the Three Gorges Reservoir area can be attributed to the unusually low winter temperature, which leads to early senescence of fruit and triggers a series of physiological, biochemical, and molecular biological processes of granulation. In the fruit-maturation stage, when freezing injury occurs, overproduction of reactive oxygen species (ROS) will result in oxidative stress, causing lipid peroxidation and membrane damage [32].

With the excessive accumulation of ROS, the membrane structure will be destroyed, and the soluble sugars and organic acids will flow out and cause fruit granulation (Figure 4F). Moreover, after freezing injury in winter and before the sprouting of the tree in the following spring, the water in the fruit that remained on the tree does not dissipate and could basically be preserved, maintaining a dynamic balance of water content in the fruit. However, the water content in the fruit is continuously lost at the germination, budding, flowering, and shoot growth stages (Figure 4B). Especially at the flowering stage, the water absorption and retention capacity of fruit rapidly decline, resulting in granulation. Besides, the tender buds, flowers, and new shoots compete for water and nutrition with the fruit remained on the tree. A series of dynamic physiological and biochemical changes occur in the remained fruit, such as accelerated respiratory metabolism. Sugars and acids in the fruit are consumed [9,12,18,33], osmotic potential is enhanced and water is transferred from the middle column, leading to the senescence of juice sac and consequently the occurrence of granulation.

However, the granulation of citrus fruit during postharvest storage is mainly caused by the senescence of the fruit and environmental factors. The loss of some organic substances in the juice sac, such as sugars and acids involved in the construction of cell wall of juice cells, results in fruit granulation [33]. A low water content will lead to increases in the respiration intensity and internal physiological consumption of the fruit, as well as significant decreases in sugars, organic acids, vitamins, and dry matter in juice cells, especially soluble sugars and organic acids [25,30]. With the extension of storage time and under the regulation of endogenous hormones and enzymes, a series of physiological and biochemical changes occur in the fruit, and the internal metabolism regulates the accumulation, transfer, and consumption of inclusions in the process of tissue senescence, leading to juice sac granulation [34,35].

## 4. Conclusions

In conclusion, low winter temperature promotes the granulation of late-ripening citrus fruit. On the other hand, RNA-Sequencing analysis showed that there were significant changes in metabolism and molecular pathway during the granulation process of late-ripening citrus fruit. In particular, this study showed that low temperature may cause the granulation of citrus fruit by affecting cell wall metabolism.

## 5. Materials and Methods

### 5.1. Materials and Sample Collection

The fruit samples were collected from Lanelate navel orange trees (*C. sinensis* Osbeck) located in three orchards of each about 10 acres at the altitudes of 520 m, 220 m, and 380 m (as control) in Zigui County, Hubei Province, in the Three Gorges Reservoir area (E 110°41′, N 30°54′). The trees were 8-year old with high-grafting Lanelate navel orange, using Robertson navel orange as intermediate rootstock and red orange as base rootstock. The soil of the orchard was sandy loam with a pH value of about 6.5 and all trees were under conventional management and medium fertility.

In this study, based on the meteorological data of 2008–2018 collected from meteorological observation station in Zigui County, the relationship between low temperature in winter and granulation of navel orange fruit in Lanelate was analyzed (Appendix A). In January and February, temperature recorders (ZDR-20) were placed at fixed points in orchards at different altitudes to automatically record the temperature, compare the difference of temperature in winter in each orchard, and analyze the impact of altitude and temperature on granulation of fruit. The orchard selected at 520 m altitude suffered from extremely low temperature of −4.1 °C for about 7 days in winter, and the extremely low temperature in the orchards at 380 m and 220 m (as control) was, respectively, −2.8 °C and −2.0 °C from January to February (Appendix A). The sampling data ranged from December 2013 to May 2014. Fruit samples were collected at 235, 250, 265, 300, 335, 350, 365, and 400 days after flowering (DAF), and 10 citrus fruits were collected from three individual trees to be used as one biological replicate. Thus, a total of 30 citrus fruit were collected for each sample, and then immediately taken to the laboratory. The picked fruit were transacted, granulated, and the normal juice sacs were separated. Fresh experimental samples were used for evaluating fruit quality, such as determination of soluble solid content (SSC), titratable acid (TA), nd cell wall components (cellulose, lignin, pectin). For tissue section assays, juice sacs with different granulation levels were chosen carefully, and then fixed in formalin–acetic acid–alcohol (FAA) fixative until use. Additional selected juice sacs, including granulated parts and normal sections, were quickly frozen in liquid nitrogen and stored at −80 °C to be used for related enzyme activity assay and RNA-sequencing. For low-temperature treatment, 100 non-disordered fruit were collected from late-ripening navel orange trees at 235 DAF, treated with 4 °C temperature, and the juice sacs were collected at 0, 3, 6, 9, 12, 15, 18, 21, and 24 h. For the nine samples, three biological replicates were analyzed in the experiments, and three citrus fruit were collected for each biological replicate. The samples were immediately put into liquid nitrogen and stored at −80 °C for future use.

### 5.2. Determination of Juice Sac Granulation Level

The degree of granulation was assessed according to the method of Wang et al. [12], Sharma et al. [11] and Zhang et al. [26] with minor modifications. Briefly, the granulation rate of fruit was expressed as the percentage of the fruit that undergo granulation in the total number of assessed fruit. The fruit was cut at the equatorial plane into equal half, and granulation was assessed by the extent of dryness on each as follows: normal, level 0; less than 25% dry area, mild, level 1; 25–50% dry area, moderate, level 2; more than 50% dry area, severe, level 3 (Figure 1). More than thirty fruits were detected in each level.

### 5.3. Fruit Quality Analysis

According to the degree of granulation, juice was prepared at each level of granulation for analysis of fruit density, water content, juice percentage, soluble solid content (SSC), and titratable acid (TA) content. Juice was prepared in the laboratory using a domestic juicer. The extracted juice was filtered through a double layer of gauze to remove seeds and albedo fragments and to reduce juice sac content. For each granulation level, 10 citrus fruit were used for juice sac preparation. For each cell wall metabolite, three biological replicates were tested. SSC was determined using a digital pocket refractometer (ATAGO, PAL-1, Tokyo, Japan). The NAOH method was used to measure the TA content [36]. The color of fruit juice was determined by a color difference meter (Minolta, CM-5, Tokyo, Japan).

### 5.4. Determination of Pectin, Cellulose, and Lignin Contents

Juice sacs were prepared at the four levels (0, 1, 2, and 3) for the determination of pectin, cellulose, and lignin contents. Ten fruits were selected from each level to separate fruit juice sacs, with three repeated determinations. Pectin was extracted from juice sacs and determined using Carbazole Colorimetric Method according to previous studies [37,38]. Cellulose and lignin contents were determined by concentrated acid hydrolysis colorimetry as described previously [39,40].

### 5.5. Cross Section Preparation and Microscopic Observeation of Juice Sacs

Microscopy was performed on fresh juice sacs according to the previously described method [39]. Granulated and normal juice sacs were immersed in FAA solution and vacuumized for 30 min. After fixation, the samples were routinely dehydrated by a conventional graded series of alcohol, and, in the end, they were transparentized in chloroform solution, embedded in paraffin, and cut into 8–10 mm slices with a microtome (Lecia RM-2255, Heidelberg, Germany). The slices were transferred onto Superfrost Plus slides and dried for 5 d at 40–45 °C, then dyed by periodic acid Schiff (PAS) staining [22], transformed into permanent slices after transparent dehydration, and observed and photographed by a digital fluorescence inverted microscope (Olympus, IX71, Tokyo, Japan).

### 5.6. RNA-Sequencing and Characterization, Sequence Assessment, and Screening of Differentially Expressed Genes (DEGs)

Juice sacs derived from late-ripening navel orange covering different granulation levels as described above were collected for subsequent RNA isolation. Total RNA was extracted from juice sacs of granulated and normal fruit, and RNA from fruit with granulation levels of 1, 2, and 3 (1:1:1) was mixed as the granulated sample. This was repeated twice, and libraries GR_1_, CK_1_, GR_2_, and CK_2_ were constructed. Total RNA was extracted from 0.2 g fresh juice sac of each sample by TRIzol reagent according to the manufacturer’s instructions (TaKaRa, Shiga, Japan).

The quantity of RNA from each sample was measured by a NanoDrop spectrophotometer and an Agilent 2100 Bioanalyzer. The RNA quality and integrity was determined by 1% gel and RIN (RNA integrity number). RNA with three clear bland in the gel and RIN value more than 7.0 and concentration ≥ 40 ng/µL were used for the following library construction. RNA library construction and sequencing were conducted at Huada Genomics Co., Ltd., Shenzhen, China. Oligo (dT) magnetic beads were used to isolate poly (A) mRNA after the 4 RNA mixtures were collected. A fragmentation buffer was added to interrupt the mRNA to produce short fragments (about 200 bp). First-strand cDNA was synthesized using random hexamer primer, and second-strand cDNA was synthesized using buffer, dNTPs, RNase H, and DNA polymerase I. The double-stranded cDNA was purified with a QIAquick PCR Purification Kit and washed with EB buffer for the end repair and poly (A) addition reaction. The cDNA fragments were purified by agarose gel electrophoresis. The cDNA libraries quality and size distribution were checked using Agilent Bioanalyzer 2100 DNA chip (Agilent Technologies Inc., Santa Clara, CA, USA). Library fragment sizes were between 200–500 bp, with a peak at approximately 260 bp. All libraries were quantified with a Qubit 2.0 Fluorometer (Life Technologies), and only cDNA libraries with a concentration more than 10 ng/µL were subjected to sequencing by Huada Genomics Co., Ltd. using an Illumina HiSeq TM 2000. Then, the raw sequence reads were deposited into a publicly available database at the National Center for Biotechnology Information Sequence Read Archive (NCBI SRA) (http://www.ncbi.nlm.nih.gov/sra) with SRA accession number: PRJNA597259.

In order to get the clear reads, the raw reads were filtered by removing the reads with adaptors, impurity raw reads and low quality reads and then, the clean reads of four libraries were mapped to reference sequences using the sweet orange genome as a reference [19]. The original image data were transferred into raw data or raw reads by base calling [41]. Then, “Clean Reads” were filtered from these raw reads by three steps: (1) removing the reads with ambiguous nucleotides or adaptor sequences; (2) discarding the reads in which unknown bases were more than 10%; and (3) removing low-quality reads. (The percentage of low-quality bases with a quality value ≤ 5 was more than 50% in a read.) Clean reads were mapped to the sweet orange genome (http://citrus.hzau.edu.cn/orange/) [19] using SOAPaligner/soap2 [42]. Mismatches no more than 2 bp were allowed in the alignment.

The expression levels of genes were calculated by using reads per kb per million reads (RPKM) [43]. The NOIseq software was utilized to identify DEGs between two samples, and a strict Poisson distribution model algorithm was performed [44,45]. Data were log_2_ transformed and filtered at 2-fold or greater differences in expression for each sample. To compare the significance differences of gene expression, the values of a FDR (False Discovery Rate) ≤ 0.001 and the absolute value of log_2_ ratio ≥ 1 were both used.

For Gene Ontology (GO) and Kyoto Encyclopedia of Genes and Genomes (KEGG) annotation, the DEGs were aligned with GO database (http://www.geneontology.org/) and KEGG database (http://www.genome.jp/kegg/), respectively. GO analysis was performed by WEGO software (http://wego.genomics.org.cn/) [46]. KEGG pathway enrichment analysis identified significantly enriched metabolic pathways or signal transduction pathways associated with DEGs compared with the whole genome background. The enrichment degree of KEGG was measured through Rich factor, Q value, and genes numbers enriched to this pathway. Pathways with corrected p-values < 0.05 were regarded as significantly enriched in DEGs. As the enriched pathway numbers were fewer than 20, all of them were put into the plot. The bar chart was used to illustrate the DEGs-enriched GO terms with R package (ver.3.6.0).

### 5.7. Validation of RNA-Sequencing Data by Quantitative Real-Time RT-PCR

To verify the DEG results, qRT-PCR gene expression analysis was conducted to confirm the sequencing consistency. The total RNA of four samples (GR1, GR2, CK1, CK2) was extracted and reverse transcribed into cDNA using a PrimeScript RT-PCR Kit (TaKaRa, Japan). Twenty genes were randomly selected for validation by qRT-PCR using SYBR Premix Ex Taq (Takara, Shiga, Japan). The primers were designed by the Primer 6.0 program (PREMIER Biosoft International, Canada). Details of gene-specific primers including endogene primers are presented in Appendix A. qRT-PCR gene expression was analyzed by a StepOne Real-Time PCR System (Applied Biosystems, Forster, America) using Actin gene as the reference. The relative expression levels of target genes were presented by the 2^−^^ΔΔCT^ method [47]. The experiment was performed in triplicate for each sample.

### 5.8. Measurement of Enzyme Activity Related to Cell Wall Metabolism in Juice Sacs

The non-disordered fruit and samples with different granulation levels were quickly frozen in liquid nitrogen and stored at −80 °C. Enzyme extraction was conducted at 4 °C using the method of Lohani et al. with minor modifications [48]. A 2.0 g juice sac or 0.5 g juice sac sample was ground into fine powder in liquid nitrogen and homogenized with 9 mL of PBS buffer (pH = 7.8) in ice bath. Then, the mixture was centrifuged at 5000× *g* for 30 min at 4 °C. The supernatant was used for determination of enzyme activity as soon as possible after extraction. The activities of CL, PG, and PME and the antioxidant enzymes SOD, POD, and CAT were determined by the corresponding enzyme linked immunosorbent assay kits (Shanghai Enzyme-linked Biotechnology, China) according to the manufacturer’s manuals. The enzyme activity was determined in units: one unit (U) of enzyme was defined as the amount that catalyzes the formation of 1 μg reducing groups per minute per gram of original fresh weight of fruit.

### 5.9. Gene Expression in Juice Sacs Treated at 4 °C by RT-PCR

Total RNA was isolated from the fruit juice sacs using the TRIzol DNA-free Kit^TM^ (TaKaRa, Shiga, Japan) according to the manufacturer’s instructions. The RNA quality and integrity was determined by 1% gel and RIN (RNA integrity number). RNA with three clear bland in the gel and RIN value more than 7.0 and concentration ≥ 40ng/μL were used for the following library construction. Reverse transcription was performed with 2 μg of total RNA using a reverse transcription kit (Toyobo). Pairs of primers for the PG, PME, CL, PAL, CAD, and POD genes were designed by Primer Express 5.0 software according to the conservative regions of amino acid sequences in GenBank. Actin was used as the internal reference gene to normalize the target gene expression levels of samples. Details of gene-specific primers including endogen primers are provided in Appendix A. Real-time PCR was performed using SYBR Green Real-time PCR Master Mix System with the following conditions: SYBR Green mix 5 μL, Primer F/R 0.5 μL/0.5 μL, cDNA 0.5 μL, and nuclear-free water 3.5 μL. Gene expression values were calculated by multiplying the signal intensity obtained. All RT-PCR reactions were performed at least in triplicate, and the mean relative expression values and standard errors were calculated.

### 5.10. Statistical Analysis

All data were evaluated by Duncan’s multiple range tests in the ANOVA program of SAS (SAS Institute, Cary, NC, USA); differences were considered to be significant at *p* ≤ 0.05.

## Figures and Tables

**Figure 1 plants-09-00095-f001:**
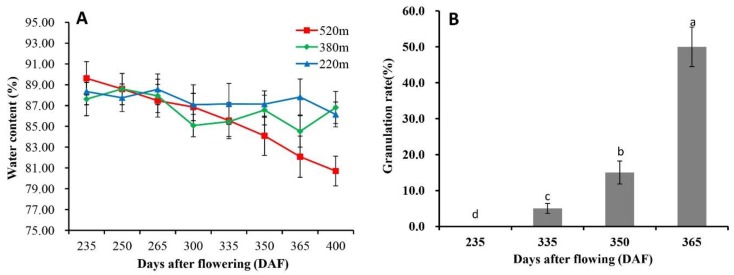
Water content in juice sac and granulation rate of late-ripening navel orange at different altitudes and harvest time. (**A**) Changes in water content in the juice sac of late-ripening navel orange at different altitudes and harvest time. (**B**) Occurrence of fruit granulation at 520 m altitude and different stages after flowering. Letters mean significant differences at *p* ≤ 0.05 level using Duncan’s multiple range test. Data are mean ± SD of three replicates. DAF, days after flowering.

**Figure 2 plants-09-00095-f002:**
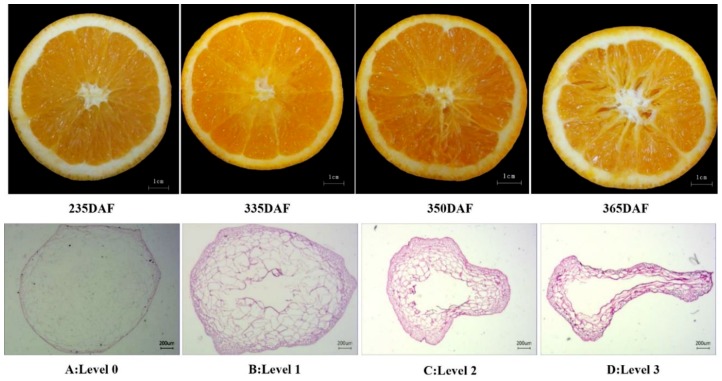
Morphology and tissue sections at different stages after flowering of late-ripening navel orange. (**A**) 235 DAF, normal, Level 0; (**B**) 335 DAF, slight, level 1; (**C**) 350 DAF, moderate, level 2; and (**D**) 365 DAF, severe, level 3.

**Figure 3 plants-09-00095-f003:**
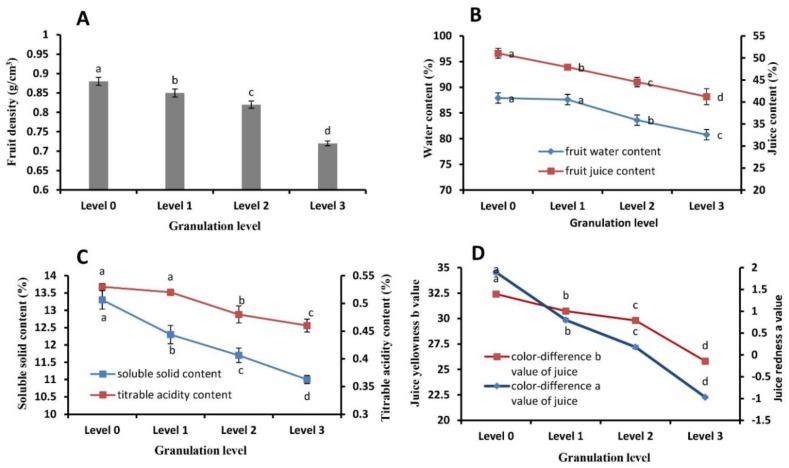
Fruit quality analysis in juice sacs of different granulation levels. (**A**) Fruit density; (**B**) fruit water and juice content; (**C**) soluble solid and titrable acidity content; (**D**) color difference value (a, red color; b, yellow color) of juice. Data are mean ± SD of three replicates.

**Figure 4 plants-09-00095-f004:**
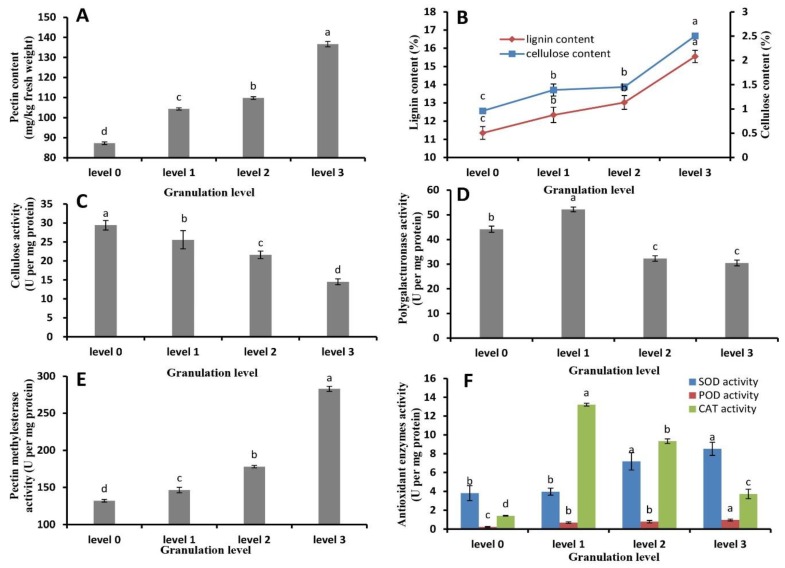
Pectin, cellulose, and lignin content and metabolic enzyme activity related to cell wall metabolism in juice sacs with different levels of granulation. (**A**) Pectin content; (**B**) lignin and cellulose content; (**C**) cellulase activity; (**D**) polygalacturonase activity; (**E**) pectin methylesterase activity; (**F**) antioxidant enzyme (superoxide dismutase (SOD), peroxidase (POD), catalase (CAT)) activity. Data are mean ± SD of three replicates.

**Figure 5 plants-09-00095-f005:**
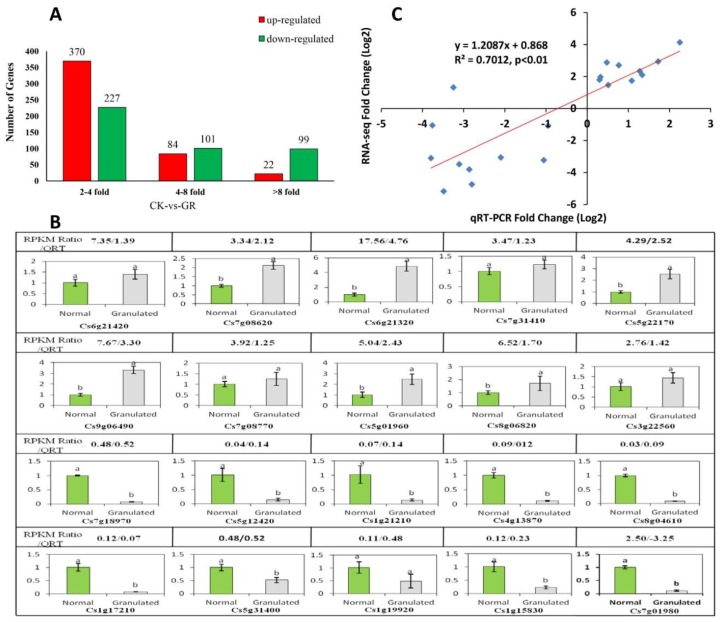
Screening of differentially expressed genes (DEGs) based on RNA-sequencing results and real-time fluorescence quantitative PCR verification between granulated fruit and non-disordered fruit of Lanelate sweet orange. (**A**) Overall numbers of DEGs between granulated fruit and non-disordered fruit of Lanelate sweet orange. (**B**) Real-time PCR confirmation of DEGs between normal fruit (green columns) and granulated fruit (grey columns). Transcript abundance (GR/CK reads per kilobase per million mapped reads (RPKM) ratio) is shown on the top of each gene. Relative transcript levels were calculated by real-time PCR with Actin as the standard. QRT refers to relative expression of genes between granulated fruit versus non-disordered fruit (2^−^^ΔΔCT^). Data are mean ± SD of three replicates. (**C**) Real-time quantitative RT-PCR confirmed 20 DEGs. Scatter plot shows the expression changes (log_2_ fold) measured by RNA-sequencing and qRT-PCR analysis of selected genes. The results are plotted for the genes that showed significantly (*p* ≤ 0.05 using Duncan’s multiple range test) up- or down-regulation. A linear trend line is shown.

**Figure 6 plants-09-00095-f006:**
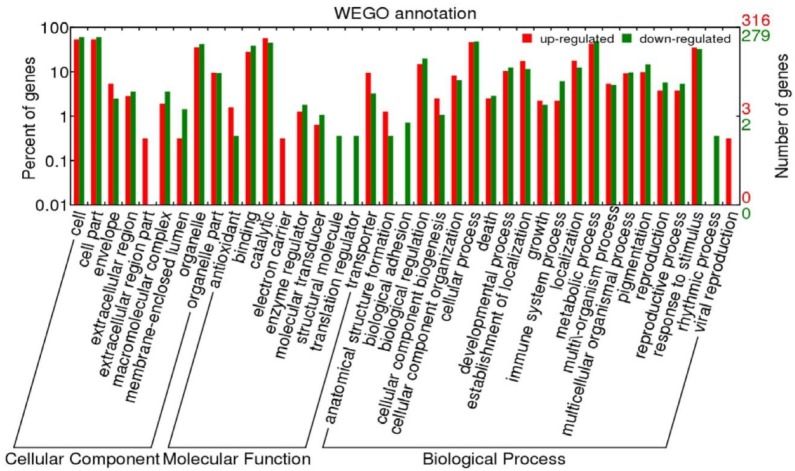
Gene Ontology (GO) assignment of DEGs. Red and green bars show the GO distributions of up-regulated and down-regulated DEGs, respectively.

**Figure 7 plants-09-00095-f007:**
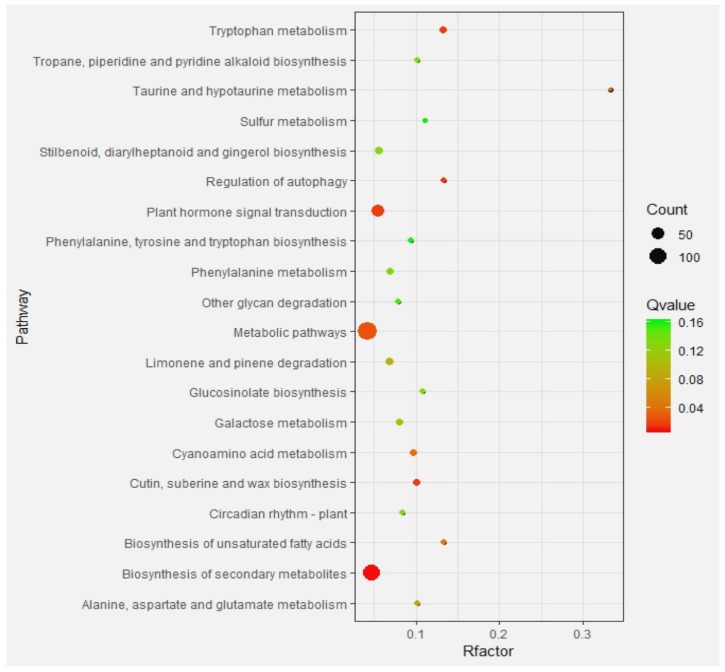
Kyoto Encyclopedia of Genes and Genomes (KEGG) pathway enrichment scatter plot of DEGs. Y-axis shows pathway name and x-axis shows Rfactor (Rich factor). Dot size represents the number of DEGs and the color indicates q-value.

**Figure 8 plants-09-00095-f008:**
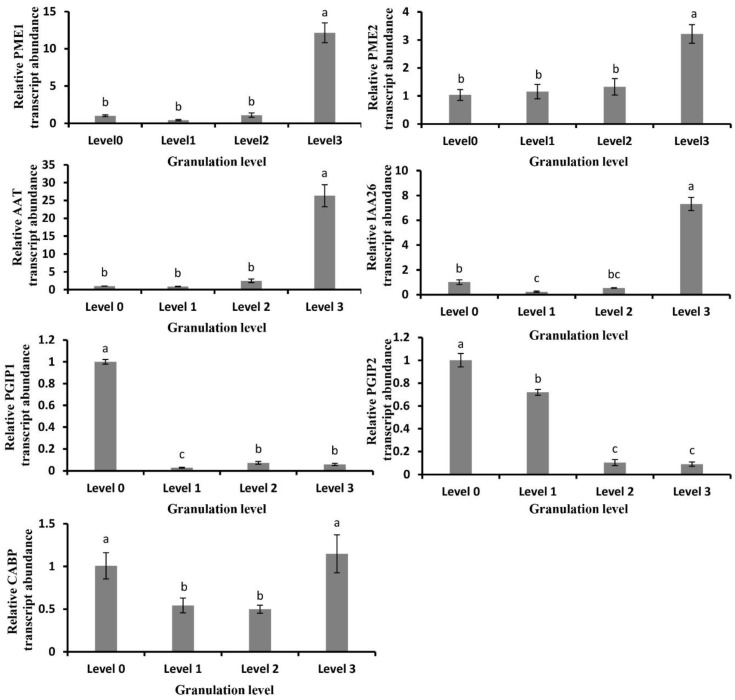
Changes of some typical DEGs related to cell wall metabolism at different granulation levels. PME1, pectin methylesterase gene 1 (orange 1.1t00214); PME2, pectin methylesterase gene 2 (*Cs*2g07660); AAT, alcohol acyl transferase gene (*Cs*2g31410); IAA26, auxin-responsive protein gene like (*Cs*1g15830); PGIP 1, polygalacturonase inhibitor protein gene 1 (*Cs*7g18970); PGIP 2, polygalacturonase inhibitor protein gene 2 (*Cs*7g01980); CABP, calcium-binding protein gene (*Cs*6g21420). The relative gene transcript abundance was calculated by 2^−^^ΔΔCT^. Data are mean ± SD of three separate determinations of the same sample.

**Figure 9 plants-09-00095-f009:**
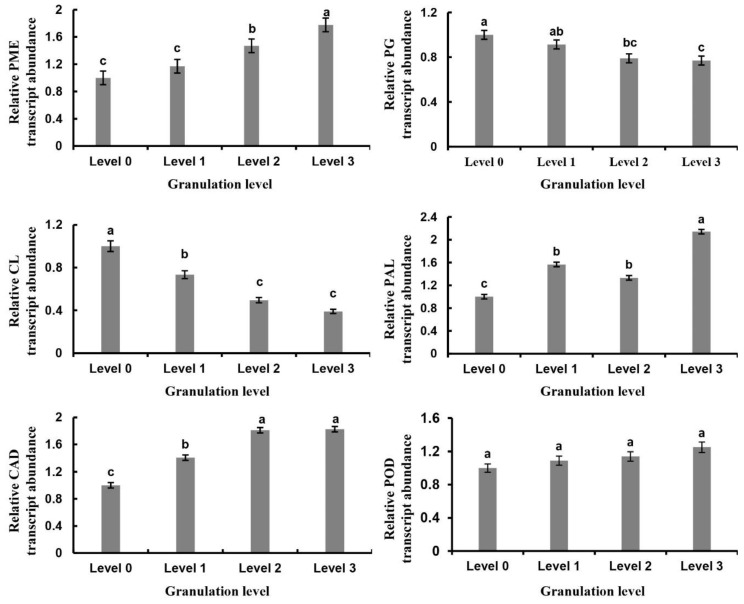
Relative expression of genes related to cell wall metabolism at different levels of juice sac granulation in late-ripening navel orange. *PME* (*Cs*1g16550); *PG* (*Cs*1g12840); cellulose (*CL*) (*Cs*5g20320); cinnamol dehydrogenase (*CAD*) (*Cs*1g04910); phenylalanine ammonia lyase (*PAL*) *(Cs*8g16290); *POD* (Orange1.1t02040). Data are mean ± SD of three replicates.

**Figure 10 plants-09-00095-f010:**
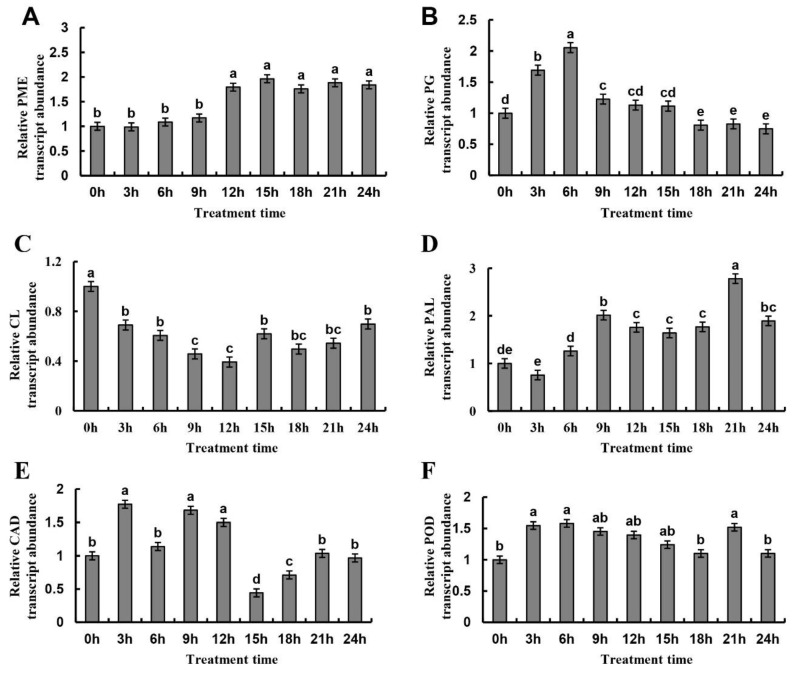
Relative expression of genes related to cell wall metabolism in the juice sacs of late-ripening navel orange under 4 °C cold stress at different time points. (**A**) Relative expression of *PME* (*Cs*1g16550); (**B**) Relative expression of *PG* (*Cs*1g12840); (**C**) Relative expression of *CL* (*Cs*5g20320); (**D**) Relative expression of *PAL* (*Cs*8g16290); (**E**) *CAD* (*Cs*1g04910); (**F**) Relative expression of *POD* (Orange1.1t02040). Data are mean ± SD of three replicates.

**Table 1 plants-09-00095-t001:** List of differentially expressed genes related to fruit juice sac granulation of late navel orange by RNA-sequencing.

Gene ID.	E-Value	RPKM Value		Log2 Fold Change	Functional Description
		CK	GR		
Cs1g04910	3.75 × 10^−77^	0.35	0.90	1.35	Cinnamyl-alcohol dehydrogenase
Cs1g12840	1.36 × 10^−49^	0.80	1.75	1.13	Polygalacturonase
Cs1g15830	6.71 × 10^−27^	14.35	1.72	−3.06	Auxin-responsive protein IAA26-like
Cs1g16550	1.26 × 10^−176^	101.57	218.40	1.10	Pectin methylesterase
Cs1g17210	2.69 × 10^−20^	34.70	4.05	−3.10	Protein TIFY 10A
Cs1g19920	0	60.62	6.48	−3.23	Cell wall protein PRY3
Cs1g21210	1.26 × 10^−114^	3.44	0.25	−3.81	1-Aminocyclopropane-1-carboxylate synthase, aminotransferase
Cs2g07660	3.25 × 10^−110^	63.99	3.19	−4.32	Pectin methylesterase
Cs2g31410	0	11.99	0.11	−6.74	Alcohol acyl transferase
Cs3g22560	0	3.26	9.00	1.47	Citrus sucrose transporter 1
Cs4g13870	4.29 × 10^−128^	562.74	50.38	−3.48	1-Aminocyclopropane-1-carboxylate oxidase
Cs5g01960	2.28 × 10^−95^	1.20	6.04	2.33	Cellulose synthase-like protein G3-like
Cs5g12420	9.56 × 10^−160^	64.54	2.42	−4.74	Epidermis-specific secreted glycoprotein EP1-like protein
Cs5g20320	5.70 × 10^−90^	0.64	0.02	−4.70	Basic cellulase
Cs5g22170	5.01 × 10^−130^	4.62	19.80	2.10	1-Associated receptor kinase 1 precursor
Cs5g31400	1.82 × 10^−100^	22.66	10.79	−1.07	Myc-like proanthocyanidin regulatory protein
Cs6g21320	3.94 × 10^−143^	0.16	2.90	4.13	Amino acid transporter
Cs6g21420	6.71 × 10^−51^	8.02	58.95	2.88	Calcium-binding protein
Cs7g01980	1.56 × 10^−147^	5.12	12.78	1.32	Polygalacturonase-inhibiting protein
Cs7g08620	0	61.27	204.54	1.74	Probable polygalacturonase non-catalytic subunit JP630-like
Cs7g08770	2.27 × 10^−92^	1.65	6.47	1.97	Galactinol synthase
Cs7g18970	7.68 × 10^−180^	1121.82	542.19	−1.05	Polygalacturonase-inhibiting protein
Cs7g31410	2.45 × 10^−111^	28.17	97.83	1.80	plasma membrane intrinsic protein
Cs8g04610	0	17.72	0.49	−5.17	GH3 family protein
Cs8g06820	0	0.54	3.50	2.70	Mannitol transporter
Cs8g16290	0	0.48	2.12	2.13	Phenylalanine ammonia-lyase
Cs9g06490	1.53 × 10^−113^	5.29	40.58	2.94	Carboxylesterase
orange1.1t00214	3.45 × 10^−149^	2.40	0.00	−7.91	Pectin methylesterase
orange1.1t02040	5.65 × 10^−^^58^	0.04	0.11	1.45	Peroxidase

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
