# Peer review of "Transcriptome Analysis Unravels Metabolic and Molecular Pathways Related to Fruit Sac Granulation in a Late-Ripening Navel Orange (Citrus sinensis Osbeck)"

_plants, 2020, doi:10.3390/plants9010095_

Round 1
Reviewer 1 Report
The manuscript Transcriptome Analysis Unravels Metabolic and Molecular Pathways Related to Fruit Sac Granulation in a Late-ripening Navel Orange (Citrus sinensis Osbeck) is very interesting. The RNA-seq is a very powerful approach for identifying differentially expression genes. I really appreciate it. Globally, the manuscript was well prepared. However I have some comments on how the authors proceed on the methods. Specific comments Introduction 1- Lines 57-58: Please provide the reference of your previous study. Materials and Methods 1- The authors should provide details about libraries quality check 2- Details about the method used for quantify the cDNA libraries must provided. This is an important step before proceeding to the sequencing. 3- information about the software/program used for reads quality check is missing. Please provide details. 4- Authors should provide details about the program used for genes expression levels and DEGs identification. 5- I recommend adding some results about the transcription factors (TFs) that were differentially expressed. Results The results were well prepared. Discussion The results were well discussed.Author Response
1.Comments and Suggestions for Authors
This is a very important research for the cultivation of Navel Orange. This manuscript is well written with a lot of data presented and interesting to read, but needs some modifications.The major issue of this manuscript is the extension of the results section compared to the rest of the sections, and especially to the discussion. Please, check the following minor issues that should be resolved.
Introduction
It’s well written and it’s an exhausted introduction and highlight very well the importance of this study. It’s very clear the purpose of the work and its significance, and the current state of the research is reviewed in an exhaustive manner.
Results
Are presented and interpreted in a concise and precise way.
Discussion
Discussion is analytic, logical, and comprehensive. However, the amount of the discussion is not proportional to the rest of the text, so it is recommended to extend the section of the discussion. Moreover, throughout the text the authors use the term ‘speculate’ several times, so please change with another one. It doesn’t sound so good.
Finally, there are some parts of the discussion without references, for example P 13, L 342, L 349.
Response: Thanks a lot. We have changed the term “speculate” in to different word according to different context in the manuscript. The missing references needed in the manuscript were also added in the reference section. However, due to the great length of this manuscript, it is not appropriate to extend our discussion section. Additionally, we improved some small parts of our discussion, we hope the new version of our discussion can meet your requirements.
Yao S, Cao Q, Xie J, et al. Alteration of sugar and organic acid metabolism in postharvest granulation of Ponkan fruit revealed by transcriptome profiling. Postharvest biology and technology, 2018, 139: 2-11.
Sheng L, Shen D, Yang W, et al. GABA pathway rate-limit citrate degradation in postharvest citrus fruit evidence from HB Pumelo (Citrus grandis)× Fairchild (Citrus reticulata) hybrid population. Journal of agricultural and food chemistry, 2017, 65(8): 1669-1676.
Ding Y, Chang J, Ma Q, et al. Network analysis of postharvest senescence process in citrus fruits revealed by transcriptomic and metabolomic profiling. Plant physiology, 2015, 168(1): 357-376.
Materials and Methods
These are described with sufficient detail to allow others to replicate and build on published results but is better to be presented before the discussion.
Also some errors presented:
P14, L394, L 395: (…made in Japan)
P 14, L 410: (Japan)
Please use a common form to present the construction country.
Response: The errors were corrected in a common form, such as (ATAGO, PAL-1, Japan) (Minolta, CM-5, Japan) (Olympus, IX71, Japan). See details in P14 Line 394, 395 and 409.
Appendix S
Table S1 and Table S2: please, check the unit. This is not present, if printed.
Table S2: Different character is used.
Table S3: Please, design some lines between the columns (example: between the different altitudes). It’s difficult to study this table.
Response: The unit for Table S1 and Table S2 was added, and the character was corrected in the new manuscript. In addition, we separated the Table S3 will lines to make it more friendly to the readers.
Conclusions
It’s better to be presented in a separate section.
Response: Ok, we have separated the conclusion in a new section named conclusion.
References
Please, extend this section, using the most recent references, if it’s possible. References should be presented in a correct way. Example: P 165, L 4: Two dots are presented.
Response: Thanks for your kind suggestion. Several most recent references were added in this section, and the form of all references were corrected.
Reviewer 2 Report
This is a very important research for the cultivation of Navel Orange. This manuscript is well written with a lot of data presented and interesting to read, but needs some modifications.
The major issue of this manuscript is the extension of the results section compared to the rest of the sections, and especially to the discussion. Please, check the following minor issues that should be resolved.
Introduction
It’s well written and it’s an exhausted introduction and highlight very well the importance of this study. It’s very clear the purpose of the work and its significance, and the current state of the research is reviewed in an exhaustive manner.
Results
Are presented and interpreted in a concise and precise way.
Discussion
Discussion is analytic, logical, and comprehensive. However, the amount of the discussion is not proportional to the rest of the text, so it is recommended to extend the section of the discussion. Moreover, throughout the text the authors use the term ‘speculate’ several times, so please change with another one. It doesn’t sound so good.
Finally, there are some parts of the discussion without references, for example P 13, L 342, L 349.
Materials and Methods
These are described with sufficient detail to allow others to replicate and build on published results but is better to be presented before the discussion.
Also some errors presented:
P14, L394, L 395: (…made in Japan)
P 14, L 410: (Japan)
Please use a common form to present the construction country.
Appendix S
Table S1 and Table S2: please, check the unit. This is not present, if printed.
Table S2: Different character is used.
Table S3: Please, design some lines between the columns (example: between the different altitudes). It’s difficult to study this table.
Conclusions
It’s better to be presented in a separate section.
References
Please, extend this section, using the most recent references, if it’s possible. References should be presented in a correct way. Example: P 165, L 4: Two dots are presented.
Author Response
Comments and Suggestions for Authors
This is a very important research for the cultivation of Navel Orange. This manuscript is well written with a lot of data presented and interesting to read, but needs some modifications.The major issue of this manuscript is the extension of the results section compared to the rest of the sections, and especially to the discussion. Please, check the following minor issues that should be resolved.
Introduction
It’s well written and it’s an exhausted introduction and highlight very well the importance of this study. It’s very clear the purpose of the work and its significance, and the current state of the research is reviewed in an exhaustive manner.
Results
Are presented and interpreted in a concise and precise way.
Discussion
Discussion is analytic, logical, and comprehensive. However, the amount of the discussion is not proportional to the rest of the text, so it is recommended to extend the section of the discussion. Moreover, throughout the text the authors use the term ‘speculate’ several times, so please change with another one. It doesn’t sound so good.
Finally, there are some parts of the discussion without references, for example P 13, L 342, L 349.
Response: Thanks a lot. We have changed the term “speculate” in to different word according to different context in the manuscript. The missing references needed in the manuscript were also added in the reference section. However, due to the great length of this manuscript, it is not appropriate to extend our discussion section. Additionally, we improved some small parts of our discussion, we hope the new version of our discussion can meet your requirements.
Yao S, Cao Q, Xie J, et al. Alteration of sugar and organic acid metabolism in postharvest granulation of Ponkan fruit revealed by transcriptome profiling. Postharvest biology and technology, 2018, 139: 2-11.
Sheng L, Shen D, Yang W, et al. GABA pathway rate-limit citrate degradation in postharvest citrus fruit evidence from HB Pumelo (Citrus grandis)× Fairchild (Citrus reticulata) hybrid population. Journal of agricultural and food chemistry, 2017, 65(8): 1669-1676.
Ding Y, Chang J, Ma Q, et al. Network analysis of postharvest senescence process in citrus fruits revealed by transcriptomic and metabolomic profiling. Plant physiology, 2015, 168(1): 357-376.
Materials and Methods
These are described with sufficient detail to allow others to replicate and build on published results but is better to be presented before the discussion.
Also some errors presented:
P14, L394, L 395: (…made in Japan)
P 14, L 410: (Japan)
Please use a common form to present the construction country.
Response: The errors were corrected in a common form, such as (ATAGO, PAL-1, Japan) (Minolta, CM-5, Japan) (Olympus, IX71, Japan). See details in P14 Line 394, 395 and 409.
Appendix S
Table S1 and Table S2: please, check the unit. This is not present, if printed.
Table S2: Different character is used.
Table S3: Please, design some lines between the columns (example: between the different altitudes). It’s difficult to study this table.
Response: The unit for Table S1 and Table S2 was added, and the character was corrected in the new manuscript. In addition, we separated the Table S3 will lines to make it more friendly to the readers.
Conclusions
It’s better to be presented in a separate section.
Response: Ok, we have separated the conclusion in a new section named conclusion.
References
Please, extend this section, using the most recent references, if it’s possible. References should be presented in a correct way. Example: P 165, L 4: Two dots are presented.
Response: Thanks for your kind suggestion. Several most recent references were added in this section, and the form of all references were corrected.